

# Genome-wide association studies reveal potential candidate genes associated with amino acid in tea plants

Qidi Wu[1,2], Li Song[1], Dingchen Bai[3], Yihan Wang[4], Yuting OuYang[3], Kaixin Rao[3], Qinfei Song[3], Juanying Wang[1], Suzhen Niu[1,2] and Yujie Ai[3]

[1] College of Life Sciences/Institute of Agro-Bioengineering, Guizhou University, Guiyang, Guizhou, China
[2] Key Laboratory of Plant Resource Conservation and Germplasm Innovation in Mountainous Region (Ministry of Education), Guizhou University, Guiyang, Guizhou, China
[3] College of Tea Science, Guizhou University, Guiyang, Guizhou, China
[4] Department of Tea Studies (Guizhou Duyun Maojian Tea College), Guizhou Vocational College of Economics and Business, Duyun, Guizhou, China

## ABSTRACT

**Background**. Amino acids, as the main flavor substances of umami in tea, are also the primary components determining the taste of tea, which is positively correlated with the quality and grade of tea. The Guizhou Plateau is located in the core area of the origin of the tea plant and has abundant tea germplasm. However, there are relatively few studies using genome-wide association studies (GWAS) to mine genes related to amino acid content in tea plants in the Guizhou Plateau.

**Results**. In this study, 78,819 high-quality single nucleotide polymorphisms (SNPs) markers were identified from 212 tea accessions composed by our group in the previous study by genotyping sequencing technology (GBS), and the population structure, genetic diversity, and GWAS of 212 tea accessions resources of tea were analysed. Phylogenetic tree and population structure analysis divided all germplasm into four inferred groups (Q1, Q2, Q3, Q4). By analysing the eight SNPs associated with amino acids obtained by GWAS, four candidate genes that may be related to amino acids were identified. Reverse transcription quantitative polymerase chain reaction (RT-qPCR) was used to verify the expression levels of four candidate genes, suggesting that there may be a potential gene that is important for the accumulation of amino acid content.

**Conclusions**. This study provides new information for the in-depth analysis of the genetic mechanism of amino acid content in tea plants and provides important genetic resources for accelerating the cultivation of new tea varieties with suitable amino acid content.

# INTRODUCTION

The tea plant is a perennial, evergreen, woody plant, and tea beverages processed from the young leaves or shoots of tea plants have become one of the most popular beverages in the world after water (*Wu et al., 2020*). Tea contains 20 protein amino acids (PAAs) and six non-protein amino acids (PAAs), among which theanine (Thea), glutamic acid

Corresponding authors
Suzhen Niu, niusuzhen@163.com
Yujie Ai, cusboaiyujie@126.com

(Glu), proline (Pro), arginine (Arg), and aspartic acid (Asp) are important amino acids that play an important role in tea quality (*Guo et al., 2023*; *Zhang et al., 2024*). For example, some amino acids exhibit certain good aromas while also giving tea characteristic umami profiles. The main component that produces the umami taste of tea is the content of theanine (*Ruan, Haerdter & Gerendás, 2010*). L-theanine imparts the tea its distinctive caramel aroma, while glutamic acid and phenylalanine add floral notes to the tea's aroma (*Yu & Yang, 2020*). In addition, amino acids are not only important nitrogen components in tea plants, but also the main form of nitrogen transport between different organs through xylem and phloem (*Besnard et al., 2016*). Therefore, mining genes related to amino acids in tea plants is of great significance for improving the quality of tea plants. The climate in Guizhou is very suitable for the growth of tea plants, which makes Guizhou rich in wild tea resources and local group varieties, which may contain a large number of resources with specific biochemical components (*Niu et al., 2019*). Although some studies have revealed the differences in amino acid content in Guizhou tea germplasm and the important factors regulating some amino acids (*Yang et al., 2022b*; *Mi et al., 2023*). However, the genetic structure and molecular basis of the natural variation of amino acid content in tea plants are still not clear. Guizhou is one of the origins of tea plants, and it is also the only province in China that has the characteristics of low latitude, high altitude, and less sunshine, which unique geographical location has created its superior natural environment (*Lee, 2017*). Previous studies have shown that within a certain altitude range, increases in altitude will enhance N metabolism in tea plants, which is conducive to the accumulation of amino acids and other compounds (*Wang et al., 2022*). At the same time, light intensity reduction is a common approach to improve the concentration of free amino acids in tea leaves (*Chen et al., 2017*). Therefore, the low latitude, high altitude, and low sunshine of Guizhou jointly affect the synthesis and accumulation of amino acids in tea plants by affecting the photosynthesis, temperature adaptability, and interaction with other environmental factors. Guizhou's unique environmental conditions provide the basis for the unique flavor and quality of Guizhou tea. But the mountains and rivers brought by the environment and the rugged roads have also brought obstacles to the local economic development. This also makes Guizhou still retain relatively complete tea germplasm. However, due to the fact that tea is a self-incompatible plant, the genome is highly heterozygous and difficult to assemble, and a large number of somatic mutations have accumulated due to long-term asexual reproduction (*Liu et al., 2023b*; *Li et al., 2024*). The protection and utilisation of tea germplasm resources in Guizhou Province is imminent.

Due to the lack of genetic resources for metabolic and genetic analysis, the current study mainly dissects the genetic mechanism of the target compounds in tea plants from the perspective of reverse genetics. In recent years, many genes involved in the amino acid pathway of tea plants have been revealed through reverse genetic studies, such as the CsPDX2.1 gene has the function of L-theanine hydrolase (*Fu et al., 2020*); CsMYB73 is a transcriptional repressor involved in L-theanine biosynthesis in tea plants (*Wen et al., 2020*), and CsLHT1 is able to transport a wide range of amino acids in tea plants (*Li et al., 2021a*). The above studies show that we can indirectly regulate amino acids through metabolic bypass pathways that affect amino acid synthesis. However, the potential

functional genes for the target traits affected by genetic and environmental factors are still largely unknown and unvalidated, which limits the prospects of tea breeding. Therefore, it is important to elucidate the genetic basis of amino acid pathways in tea plants and provide a useful resource for improving tea quality (*Huang et al., 2022b*).

Forward genetic methods such as quantitative trait locus (QTL) mapping and association mapping are important tools for identifying regulatory sites of free amino acid content in higher plants. *Zhong et al. (2011)* performed phenotypic determination of protein and amino acid content in recombinant inbred line populations and identified 112 rice QTLs by QTL mapping. *Xu et al. (2015)* identified 17 QTLs associated with lysine, threonine, and methionine content in rapeseed meal through QTL mapping. In tea plants, QTL mapping has been used to identify important qualities and agronomic traits. *Ma et al. (2018)* identified 10 genetic variants related to theobromine and caffeine content and proportion in tea plants through QTL mapping. *Huang et al. (2022b)* identified four QTLs associated with the content of free amino acids (theanine, glutamic acid, glutamine, aspartic acid, and arginine) in tea plants through QTL mapping. *Liu et al. (2023a)* combined QTL mapping, GWAS, and transcriptomic analysis to identify a candidate gene associated with the timing of spring bud flush in tea plants. However, accurate QTL mapping based on linkage analysis first requires the construction of a specific large mapping population, which is a very time-consuming process for tea plants with long juvenile stages, and because of the use of parental hybridisation, the allele variation captured in QTL mapping becomes limited, so that the natural variation of natural populations cannot be fully understood (*Wang et al., 2019*; *Zhang et al., 2021*). Different from QTL mapping, genome-wide association studies (GWAS) have the advantages of being rapid, inexpensive, and high-resolution and have become a powerful tool for identifying multi-related candidate genes that regulate major crop traits. Based on the principle of linkage disequilibrium (LD), GWAS analyses the relationship between genetic variation and population sample characteristics and excavates genetic loci related to target traits, so as to provide a basis for the improvement of crop target traits and a new research path and ideas for molecular breeding. At present, GWAS has been widely used in the study of metabolite traits in many model plants and widely cultivated crops, such as Arabidopsis (*Filiault & Maloof, 2012*), maize (*Wen et al., 2014*), and rice (*Sun et al., 2020*). In recent years, many researchers have used GWAS to identify candidate genes associated with important agronomic traits in tea plants. *Wang et al. (2019)* used GWAS to identify favourable single nucleotide polymorphisms (SNPs) alleles as well as candidate genes controlling *C. sinensis* TBF. *Chen et al. (2023)* obtained SNP markers and candidate genes significantly associated with four leaf traits using GWAS analysis. Based on GWAS, *Wang et al. (2024)* identified four potential candidate genes associated with OTL and OtL in tea plants.

With the rapid development of next-generation sequencing (NGS) technology, whole-genome sequencing (WGS) data based on high-throughput genotyping and millions of SNPs have been widely used in various genetic studies (*Chung et al., 2017*; *Gaudin & Desnues, 2018*; *Zhao, Wang & Yuan, 2020*). WGS is the most straightforward strategy to achieve GWAS, but it is still an expensive means for most researchers. Genotyping by sequencing (GBS) has distinct advantages over WGS, such as being fast, simple, and

low-cost. It uses restriction enzymes (RE) to digest DNA, perform high-throughput sequencing of sequences at both ends of the digestion fragment, and genotype the SNP information obtained by analysis (*He et al., 2014*). GBS technology has been widely used in the analysis of genetic diversity, genetic mapping, agronomic trait localisation, and related gene mapping of wheat (*Alipour et al., 2017*), soybean (*Sonah et al., 2015*), apple (*Lee et al., 2017*), coffee (*Hamon et al., 2017*), tea plant (*Huang et al., 2022b*; *Huang et al., 2022a*), and other crops.

In this study, 212 tea accessions from the Guizhou Plateau were used as the research objects, and their population structure, linkage disequilibrium, and genetic diversity were analyzed. At the same time, we used the phenotype of the amino acid trait and the genotype of 212 tea accessions to determine the loci related to the amino acid content by GWAS technology. The study found that there may be a candidate gene associated with amino acid content. The results of this study provide useful information for the future study of the key genes of the amino acid metabolism pathway in tea plants and are of great significance for the use of molecular breeding technology to improve the amino acid traits of tea plants.

## MATERIALS & METHODS

### Plant materials and sample preparation

The 212 tea plant materials in this study were derived from the core germplasm population constructed by the research group using 415 tea germplasm resources in the early stage (Table S1) (*Niu et al., 2019*). Among them, 210 materials were from Guizhou Province, and the remaining two materials were from Fujian Province and Zhejiang Province, respectively. The experimental materials were all planted in the teaching experimental base of Guizhou University (106°40′E and 26°20′N). Each plant material is vegetatively propagated and grown in a randomised complete block design with three replicates per plant set to ensure consistency and reliability. Protective rows were set around the experimental field, and the field management of soil fertility and irrigation in the experimental field was basically the same.

In the spring of 2020, samples were taken at the teaching experimental base of Guizhou University, and the sampling standard was one bud and two leaves. Fresh leaves harvested from each accession are immediately frozen in liquid nitrogen, freeze-dried, and stored in an ultra-low temperature freezer (−80 °C) for later use.

### Phenotypic data analysis

In this study, an ultraviolet spectrophotometer was used to detect the amino acids in the samples according to the national standard method for the determination of the total amount of free amino acids in tea (GB/T8314-2013). The detection procedure is as follows: Accurately weigh 3 grams of ground dry sample, add 450 ml of boiling water, and immediately move it into a boiling water bath for 45 min, shaking every 10 min. It is then filtered under reduced pressure while hot, and the filtrate is placed in a 500 ml volumetric flask. The above extraction steps are repeated 2–3 times, and the filtrate is combined. After the filtrate is cooled, distilled water is added to set the volume to 500 ml for later use. Aspirate 1 ml of sample solution, inject it into a 25 ml volumetric flask, add 0.5 ml of

phosphate buffer with pH of 8 and 0.5 ml of 2% ninhydrin solution, heat it in a boiling water bath for 15 min, cool it to room temperature, and then add water to set the volume to 25 ml. After 10 min, the absorbance was measured at a wavelength of 570 nm with a one cm cuvette with distilled water as a blank. The amino acid content is obtained according to the calculation formula in the national standard method.

## Genotype analysis

The sequencing data obtained in the previous research of our group were used (*Niu et al., 2019*). The raw DNA reads were demultiplexed using a barcode, and the adapter was trimmed using a custom Perl script. Only the reads with quality values > 5 are kept as clean data and then aligned to the reference genome (http://tpia.teaplants.cn/) using BWA-MEM (v. 0.7.10) with default parameters (*Xia et al., 2020a*; *Jung & Han, 2022*). GATK (v. 3.7.0) was used to call for SNPs (*McKenna et al., 2010*). SNP filtering refers to previous research methods (*Danecek et al., 2011*; *Niu et al., 2019*). The specific filtering criteria are summarised as follows: (1) Biallelic SNPs must be selected. (2) "QUAL < 50.0 || QD < 2.0 || FS > 60.0 || MQ < 40.0 || Mapping Quality Rank Sum < −12.5 || Read Pos Rank Sum < −8.0" was used in variant filtration in GATK (v 3.7.0) to filter the SNPs. (3) VCFtools (v. 0.1.15) was used to filter SNPs with minor allele frequencies (MAF) <0.05 or missing data rates >20%. After filtering, 78,819 SNPs were retained from 212 tea accessions for further analysis.

## Population structure and linkage disequilibrium

The .ped and .map files of the SNP genotype data were converted into .bed files using Plink software and used as input files. Using ADMIXTURE (v. 1.3.0) software for population structure analysis (*Alexander, Novembre & Lange, 2009*). The maximum likelihood estimation clustering method based on the Bayesian model is also applied. The number of clusters (*K*-values) is assessed by determining the cross-validation error (CV error), which is tested from two to nine and runs five iterations for each *K*-value, and the optimal *K* value was confirmed according to the minimum CV error estimated by ADMIXTURE (*Liu et al., 2020*). We set the threshold to 0.8 to distinguish between pure and mixed groups. Then the principal component analysis was carried out using TASSEL (v. 5.2.72) software, and 3D scatter plots were drawn using R (v. 4.4.1) software; each individual and population was color-labeled, and the clustering relationship between them was observed (*Glaubitz et al., 2014*). Then using MEGA (v. 10.2.4) software for Neighbor-Joining Tree mapping (*Stecher, Tamura & Kumar, 2020*). Finally, based on the allele frequency correlation ($r^2$), PopLDdecay (v. 3.30) calculates and plots the squared correlation coefficient ($r^2$) between genome-wide unpruned SNPs (*Zhang et al., 2019*). The window size for LD decay analysis is 300 kb. The $r^2$ threshold was 0.21.

## Genetic diversity analysis

Genetic diversity indicators included observed heterozygosity (*Ho*), expected heterozygosity (*He*), minor allele frequency (*MAF*), inbreeding coefficient (*Fis*), nucleotide diversity (*Pi*), Tajima's *D*, and genetic differentiation coefficient (*Fst*). Using Plink (v. 1.9), calculate the *MAF*, *Ho*, *He*, and *Fis* values for each inferred population (*Chang et al., 2015*). Using

VCFtools (v. 0.1.15) to calculate the $Pi$ and Tajima's $D$ for each inferred population and the $Fst$ value for the two pairs of the inferred population (*Danecek et al., 2011*). SPSS (v. 26, Armonk, NY, USA) statistical software counts significant differences between indicators. The larger the $Fis$, the closer the individuals in the group. When $Ho$ is greater than $He$, the population follows the Hardy–Weinberg equilibrium. Failure to do so may indicate some form of selection, drift, migration, or non-random pairing in the population, leading to deviations in genotype frequency. The larger the $Pi$, the more polymorphic this locus is in the population, with a large amount of variation. The significance level of Tajima's $D$ can tell whether a locus has undergone neutral evolution. The smaller the $MAF$, the worse the polymorphism at the SNP locus. The larger the $Fst$ value, the greater the genetic differentiation between subgroups (*Danecek et al., 2011*).

## Genome-wide association study

In order to ensure the accuracy of the results, SNPs with a minor allele frequency of less than 0.05 or a maximum deletion genotype frequency greater than 20% were screened for GWAS. Based on the 78,819 high-quality SNPs and 1 phenotypic trait obtained after GBS sequencing and filtering, six GWAS linear analysis models were performed by TASSEL (v. 5.2.72) software, which were (1) GLMQ = SNP + Q (2) GLMP = SNP + P (3) MLMQ + K = SNP + Q + K (4) MLMP + K = SNP + P + K (5) CMLMQ + K (6) CMLMP + K. These six models were compared to find the optimal model among the amino acid quality traits analysed (*Bradbury et al., 2007*). Among them, the general linear model (GLM) uses the population structure (Q) or principal component analysis (PCA) matrix as a fixed effect to control for possible false positives of traits. The mixed linear model (MLM) and the compressed mixed linear model (CMLM) use the kinship matrix (K) as a covariate to reduce the inequality correlation between genotypes (*Yu et al., 2017*). Comparing the Q–Q plots of the output of the six GWAS models, the best model fitting the curve of the expected value to the observed value was entered into the later analysis as the optimal model for the trait. Manhattan plots showed the correlation between SNP sites on each chromosome and this trait. $-\text{Log}_{10}^{(P)} \geq 4.0$ was selected as the threshold to identify SNP sites closely associated with traits (*Huang et al., 2021*).

## Identification and annotation of candidate genes

The best model for the trait was selected to screen for significant SNP loci and to search for candidate genes within 5 kb upstream and downstream of significant SNP loci. SNP marker sequences significantly associated with amino acids were used as probes, and BLAST alignment was performed in the NCBI (https://www.ncbi.nlm.nih.gov/) and TPIA (http://tpia.teaplants.cn/) databases to obtain candidate genes related to amino acids, and the candidate genes were functionally annotated (*Zou et al., 2013*).

## Gene expression analyses using RT-qPCR

The nine tea materials (including three with high amino acid content, three with moderate amino acid content, and three with low amino acid content) were selected from 212 tea accessions for RT-qPCR. Firstly, the total RNA of nine samples of tea tree materials was extracted using the RNAprep Pure Polysaccharide and Polyphenol Plant Total RNA

Extraction Kit of Beijing Tiangen Biochemical Technology Co., Ltd. Then, using RNA as a template, cDNA was synthesized using Hiscript reverse transcriptase (Novozan Biology, San Diego, CA, USA) under the following conditions: 42 °C water bath for 2 min, 37 °C for 15 min, and 85 °C for 5 s. Finally, the ChamQ Universal SYBR qPCR Master Mix Kit (Novozan Biology, San Diego, CA, USA) was used for RT-qPCR, that is, the expression levels of selected candidate genes in tea cultivars with different amino acid contents were detected using cDNA as a template. The specific reaction conditions were as follows: Pre-denaturing at 95 °C for 30 s; denaturation at 95 °C for 10 s, annealing at 60 °C for 30 s, 40 cycles. GAPDH (Forward primer: AGCTGCACAACCAACTGTTTG, Reverse primer: AGCTGCACAACCAACTGTTTG) was used as the reference gene for relative quantitative analysis. Finally, the results were analysed by the $2^{(-\Delta\Delta Ct)}$ method (*Livak & Schmittgen, 2001*). A total of three biological replicates and three technical replicates were performed. Statistical analysis was performed by one-way ANOVA.

### Statistical analyses

Origin2021 software was used to create box plots of amino acid content to identify and exclude outliers. IBM SPSS Statistics (v. 26, Armonk, NY, USA) was used to perform descriptive statistical analyses of phenotypic traits, including range, mean, standard deviation (SD), and coefficient of variation (CV) for traits.

## RESULTS

### SNP marker mining and population structure

A total of 29,393,327 SNPs were obtained from 212 tea accessions, and after strict filtering conditions, 78,819 high-quality SNPs with a minor allele frequency (MAF) of >0.05 and deletion rate <20% were retained for subsequent analysis. In addition, we further investigated the distribution of high-quality SNPs. The results showed that all high-quality SNPs were randomly distributed on 15 chromosomes, with an average number of 4,811 SNPs per chromosome. Chromosome 1 contains the largest number of high-quality SNPs (7,596 SNPs). Chromosome 15 contains the least number of high-quality SNPs (3,273 SNPs) (Fig. S1 and Table S2). The lowest (24 SNPs/Mb) and highest (34 SNPs/Mb) SNP densities were detected on chromosomes 5 and 1, respectively (Table S3).

In order to further explore the genetic relationship of 212 tea accessions, a total of 78,819 high-quality SNPs were used for population structure analysis, principal component analysis, and phylogenetic analysis. The result revealed that when $K$ increased sequentially from two to nine, the CV errors were found to increase gradually (Fig. 1A). The CV error values are 0.57840 and 0.57885 at $K = 2$ and $K = 3$, respectively (Fig. 1A). There is no significant difference between the two CV errors. In the process of the evolution of tea plants, due to long-term natural selection and artificial domestication, ancient landraces and modern landraces have gradually formed. In this study, it was observed that when $K = 2$, 212 tea accessions could be divided into ancient landraces group, wild tree group, and admixture group, while when $K = 3$, 212 tea accessions could be divided into ancient landraces group, modern landraces group, wild tree group, and admixture group (Tables S4 and S5). The above results showed that when $K = 3$, the landrace group

could be better subdivided into ancient and modern landrace groups. In order to perform more accurate grouping, we consider it possible to divide 212 tea accessions at $k = 3$. The genetic component ($Q$ value) $\geq 0.8$ indicated that the genetic background of the material was relatively single, and the genetic progenitor ($Q$ value) $< 0.8$ meant that the genetic background was relatively complex and belonged to the mixed component. In this study, the population was divided into three pure groups and one mixed group (Fig. 1B and Fig. S2). The first pure group (referred to as the 'Ancient landraces group' or 'Q1' from now on) contained 77 tea accessions, including 74 *C. sinensis* (85.14% of which are ancient landraces) and three *C. remotiserrata*. The second pure group (referred to as the 'Modern landraces group' or 'Q2' from now on) consisted of 18 *C. sinensis* (66.67% are modern landraces), four *C. tachangensis,* and one *C. remotiserrata*. The third pure group (referred to as the 'Wild tree group' or 'Q3' from now on) contained seven *C. tachangensis* and two *C. remotiserrata*. The fourth admixture group (Q4) contained 103 tea accessions, including 46 *C. sinensis* and 32 *C. remotiserrata*, 22 wild *C. tachangensis*, and three near *C. taliensis* (Table S5). PCA was performed by the first three principal components, indicating that the tested accessions separated into four subgroups (Fig. 1C). Although each subgroup has a small number of joins at the intersection position. The phylogenetic tree divides the population into four subgroups by the same SNP set (Fig. 1D), which is consistent with the results of population structure and PCA analysis.

## Genetic diversity and differentiation

Based on obtaining the high-quality SNPs, we estimated the parameters of genetic diversity (*Pi*, *Ho*, *He*, *MAF*, and *Fis*) of 212 tea accessions and compared the genetic differences among four inferred populations. The results showed that the *Pi*, *Ho*, *He*, *MAF*, and *Fis* of 212 tea accessions were 0.2166, 14.1760, 6.2592, 0.1370, and 0.6646, respectively (Fig. 2A). Comparing the correlation coefficients among the four inferred populations, the results showed that the Q2 subgroup had the highest *Ho* and *He* among all populations, and the *Ho* of all inferred populations was greater than *He*, indicating that all populations had rich population genetic diversity (*Zhang et al., 2018*; *Schmidt et al., 2021*). The Q4 subgroup had the highest levels of *Pi* and *MAF* among all groups. The Q1 subgroup had the lowest *Ho* and *He* among all groups. The Q2 subgroup had the lowest *MAF* and *Fis* among all groups. The Q3 subgroup had the lowest *Pi* among all groups but the highest Fis, indicating that the individuals in this group were closely related, and the germplasm of the population was wild-type, which also confirmed the accuracy of the above results. Previous studies showed that Tajima's $D > 0$ indicates the presence of a large number of medium-frequency alleles; this may be due to population bottleneck effects or balanced selection. *MAF* is commonly used in genetics to distinguish common polymorphisms in populations from rare variants (*Nebert & Vesell, 2013*). The *MAF* of the four tea populations was greater than 0.01 and less than 0.2, indicating that the four tea groups may be in the range of moderate frequency variation. At the same time, the Tajima's $D$ values for all four tea populations were positive, indicating that they had all experienced population bottlenecks or equilibrium selection (Fig. 2A) (*Tandoh et al., 2021*; *Pandey et al., 2021*). Among the four subgroups, the genetic differentiation coefficients (*Fst*) analysis showed that the Q1 and Q3 subgroups had the

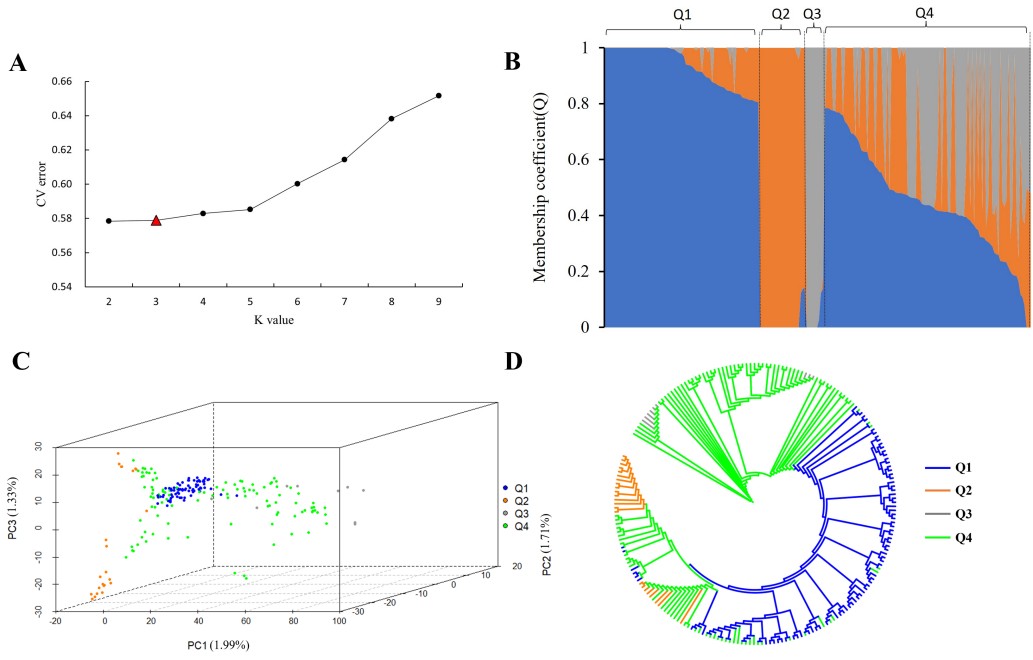

**Figure 1   SNP genotyping data were used to analyze the population structure of 212 tea accessions.**
(A) The CV error is used to estimate the tea population, and the *K* range is 2∼9. (B) Population structure based on *k* = 2–9. Q1, Q2, Q3, and Q4 represent subgroup I, subgroup II, subgroup III and subgroup IV, respectively, and blue, orange, and gray represent groups that may have come from three different ancestral groups, respectively. (C) PCA of 212 tea accessions. The first, second, and third coordinates indicate PC1, PC2, and PC3. Each dot represents an individual. Blue, orange, gray, and green, represent subgroup I (Q1), subgroup II (Q2), subgroup III (Q3), and subgroup IV (Q4), respectively; (D) Phylogenetic trees of 212 tea accessions (blue, orange, gray, and green represent subgroup I (Q1), subgroup II (Q2), subgroup III (Q3), and subgroup IV (Q4), respectively).

highest degree of differentiation, with the *Fst* value of 0.2021. The Q2 and Q4 subgroups were the least differentiated, with the *Fst* value of 0.0369 (Fig. 2A).

## Linkage disequilibrium analyses

Linkage disequilibrium (LD) decay is caused by linkage imbalance; LD decay rate varies in different species or different subspecies. LD decay is usually used to determine the general distance as the attenuation distance of the population; if the LD decay is fast, more loci are needed to achieve a certain accuracy when performing GWAS analysis (*Sallam & Martsch, 2015*). Based on the high-quality SNPs of the tea genome, we used Pop LDdecay to calculate the LD, and the resulting linkage imbalance attenuation plot showed that the LD decreased significantly with the increase of physical distance (Fig. 2B), but the average distance when the LD dropped to half of the maximum value was very short, approximately 5 kb, which indicated that the genetic diversity of the tea population was high. Therefore, we consider that 5 kb can be used as a search range of candidate genes.

## Phenotypic variation analysis of amino acid

Through the statistical analysis of the phenotypic data of 212 tea accessions in Guizhou, the results showed that the amino acid content was between 1.31% and 5.53%, the mean value

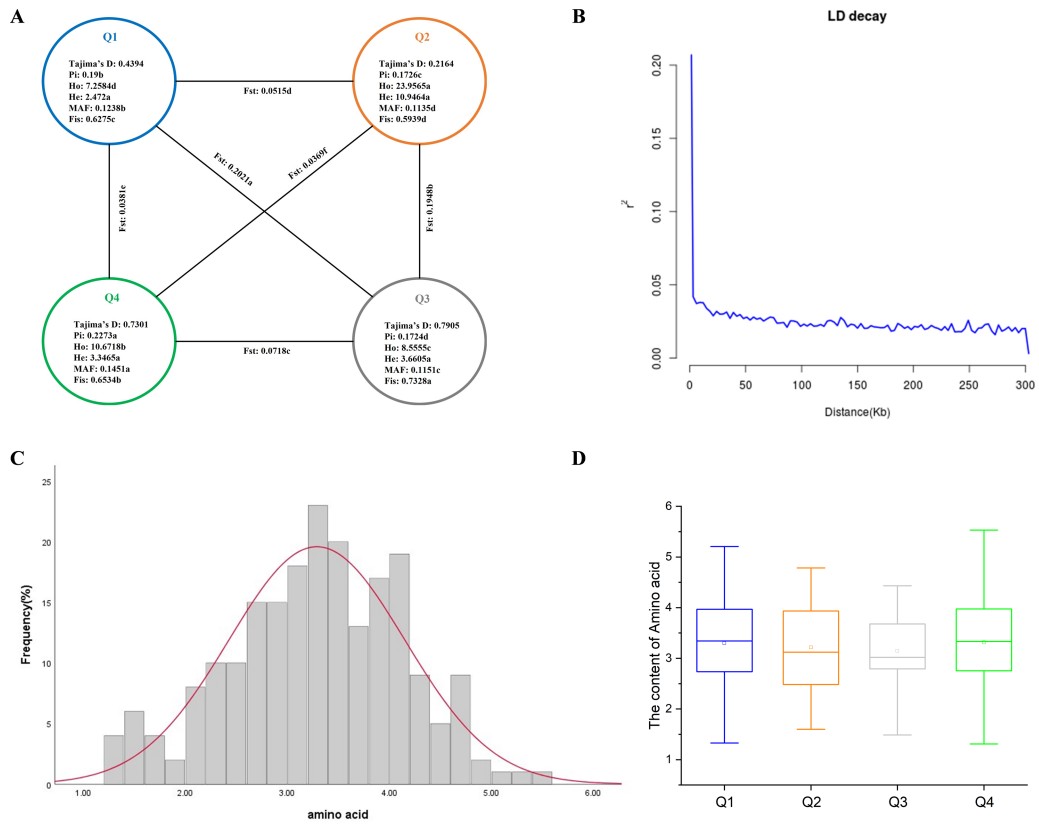

**Figure 2 Genetic diversity, LD decay analysis, and amino acid phenotype analysis of 212 tea germplasms.** (A) Genetic diversity of four inferred populations of 212 tea accessions. *Pi* nucleotide diversity, *Ho* observed heterozygosity, *He* expected heterozygosity, MAF minor allele frequency, *Fis* inbreeding coefficient, *Fst* differentiation coefficient. The different letters indicate a significant difference in $p = 0.05$ levels by the *T*-test. (B) LD attenuation plots of 212 tea accessions. (C) Frequency distribution of amino acids in environment. (D) Box plot of amino acids contents in four inferred groups.

was 3.29%, the standard deviation was 0.86, and the coefficient of variation was 26.24%. The above results showed that there was a wide variation of amino acids in the 212 tea accessions (Table 1 and Table S6). When the skewness $Z$-score and kurtosis $Z$-score values were within the range of $\pm 1.96$, the data in this group followed a normal distribution (*Orcan, 2020*). The skewness $Z$-score and kurtosis $Z$-score of amino acid content varied between $\pm 1.96$, indicating that the amino acid content was in a normal distribution (Fig. 2C), indicating that the amino acid content of 212 tea accessions was eligible for GWAS.

Amino acid content was compared between the four inferred groups. The results showed that Q4 had the highest amino acid content among the four inferred groups, followed by Q1, Q2, and Q3 (Fig. 2D). Besides, the average of amino acid content of Q1 and Q4 is comparable, but Q4 has a wider range of amino acid content.

**Table 1  Phenotypic statistical analysis of amino acids content.**

| Group | Mean (%) | Range (%) | SD | CV (%) | Skewness | Kurtosis |
|-------|----------|-----------|------|--------|----------|----------|
| Q1 | 3.30 | 1.33–5.21 | 0.86 | 26.06 | −0.219 | −0.381 |
| Q2 | 3.21 | 1.60–4.79 | 0.89 | 27.73 | 0 | −0.648 |
| Q3 | 3.15 | 1.49–4.43 | 0.91 | 28.89 | −0.424 | −0.066 |
| Q4 | 3.31 | 1.31–5.53 | 0.87 | 26.28 | −0.256 | −0.059 |
| All | 3.29 | 1.31–5.53 | 0.86 | 26.14 | −0.216 | −0.307 |

## Genome-wide association analysis

Six linear regression models (GLM-P, GLM-Q, MLM-P+K, MLM-Q+K, CMLM-P+K, and CMLM-Q+K) were used to evaluate the correlation between amino acid phenotypes and genotypes. QQ plots were used to assess the extent of accordance between the observed and expected *P*-values. Among the six models, the GLM-P was found to have the best curve fit between the expected value and the measured value, which shows that the model can be selected for subsequent analysis (Fig. 3).

For the six models, we chose $-\log_{10}^{(P)} \geq 4.0$ as the threshold for screening important loci (Fig. 4) (*Huang et al., 2021*). Eight SNPs were found to be associated with amino acids by using the GLM-P model; the contribution rate ($R^2$) of SNP phenotypic variation was 8.82% to 11.36%. The GLM-Q analysis found 15 SNPs associating with amino acids, the $R^2$ value ranging from 9.36% to 14.15%. Three SNPs were identified as associating with amino acids by using the MLM-P+K model, with the $R^2$ value ranging from 9.71% to 11.99%. The MLM-Q+K analysis found four SNPs associated with amino acids, with the $R^2$ value ranging from 9.83% to 12.73%. The CMLM-P+K and CMLM-Q+K analyses found one SNP associated with amino acids, respectively. The contribution rate ($R^2$) of two SNPs phenotypic variations was 11.85% and 11.31% (Table 2 and Table S7).

## Candidate gene screening and gene functional annotation

The genes detected in each significant locus, as well as the genes located upstream or downstream of the loci, were designated candidate genes, which were functionally annotated (*Fang et al., 2021*). Based on the LD results, 5 kb was selected as a reasonable range for searching genes. A total of four candidate genes were detected in GLM-P; all genes were subsequently functionally annotated using Blast in the TPIA (http://tpia.teaplants.cn/index.html). The *TEA022785.1* gene may have ribokinase (RBSK) activity and is involved in the synthesis of D-ribose-5-phosphate, which in turn produces ribose pyrophosphate (PRPP), which can subsequently be involved in the synthesis of histidine and tryptophan. The *TEA027937.1* gene may have L-tryptophan-pyruvate aminotransferase 1-like activity, which can catalyse the conversion of L-tryptophan and pyruvate to indole-3-pyruvate (IPA) and alanine, resulting in alanine accumulation. The *TEA013853.1* gene may have a DNA methylation 3-like factor, which may contain an XS domain, which contains a conserved aspartate residue, which may be functionally important. It may also contain an XH domain, which contains a conserved glutamate residue that may have an important function. And these two domains may interact with each other. The *TEA008491.1* gene may

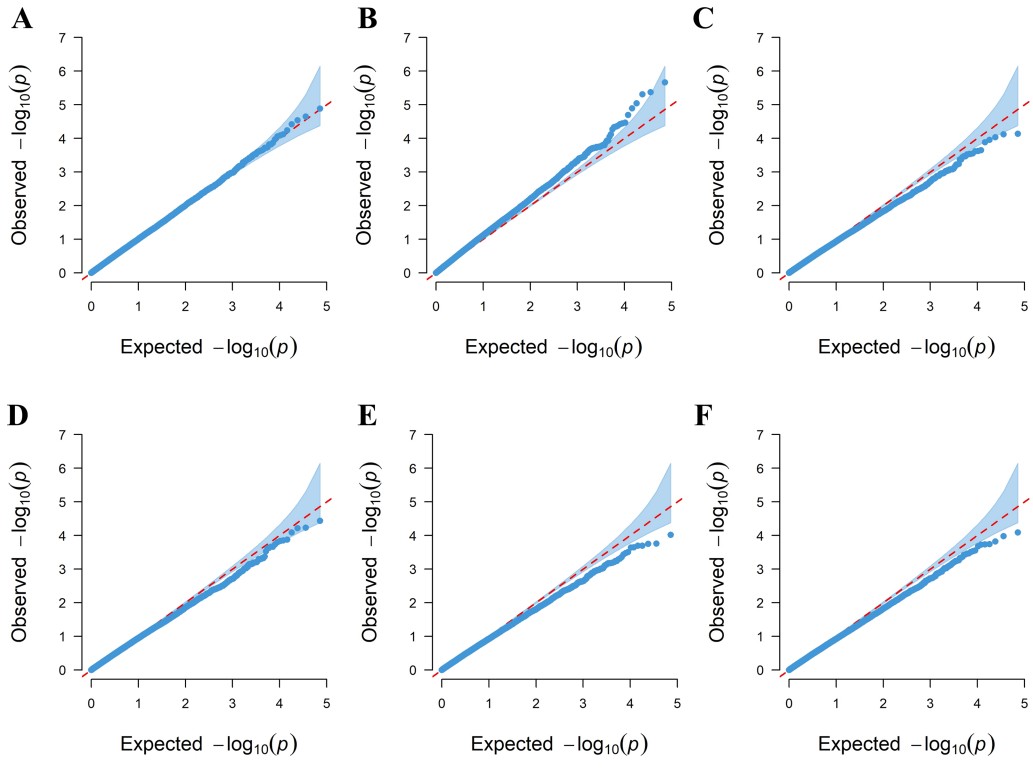

**Figure 3** **QQ plots for the amino acid traits of tea plant by six models.** (A) GLM-P; (B) GLM-Q; (C) MLM-P+K; (D) MLM-Q+K; (E) CMLM-P+K; (F) CMLM-Q+K.

have xylan α-glucuronosyltransferase 1-like activity and may be involved in the substitution of the xylan backbone in stem glucuronide ketoglycan (Table 3).

## Candidate gene validation

In order to verify whether the potential candidate genes of amino acids are involved in the accumulation of amino acid content in tea leaves, the expression levels of *TEA022785.1*, *TEA027937.1*, *TEA013853.1,* and *TEA008491.1* genes were determined by using the tea leaves of tea accessions with high amino acid content, moderate amino acid content, and low amino acid content by RT-qPCR. The results showed that the expression of the *TEA027937.1* gene was positively correlated with the amino acid content of the tea plant. The expression of the remaining *TEA022785.1*, *TEA013853.1*, and *TEA008491.1* genes was irregular (Fig. 5, Fig. S3, Tables S8 and S9). Therefore, the above *TEA027937.1* gene may be involved in regulating the accumulation of amino acid content in tea.

## DISCUSSION

### Genetic variation in tea plant populations

Tea plants originated in southwestern China and later spread around the world (*Yamanishi, 1995*). Natural selection and artificial domestication have yielded a wide variety of tea germplasm, including cultivars, landraces, and wild types (*Fang et al., 2021*). Population

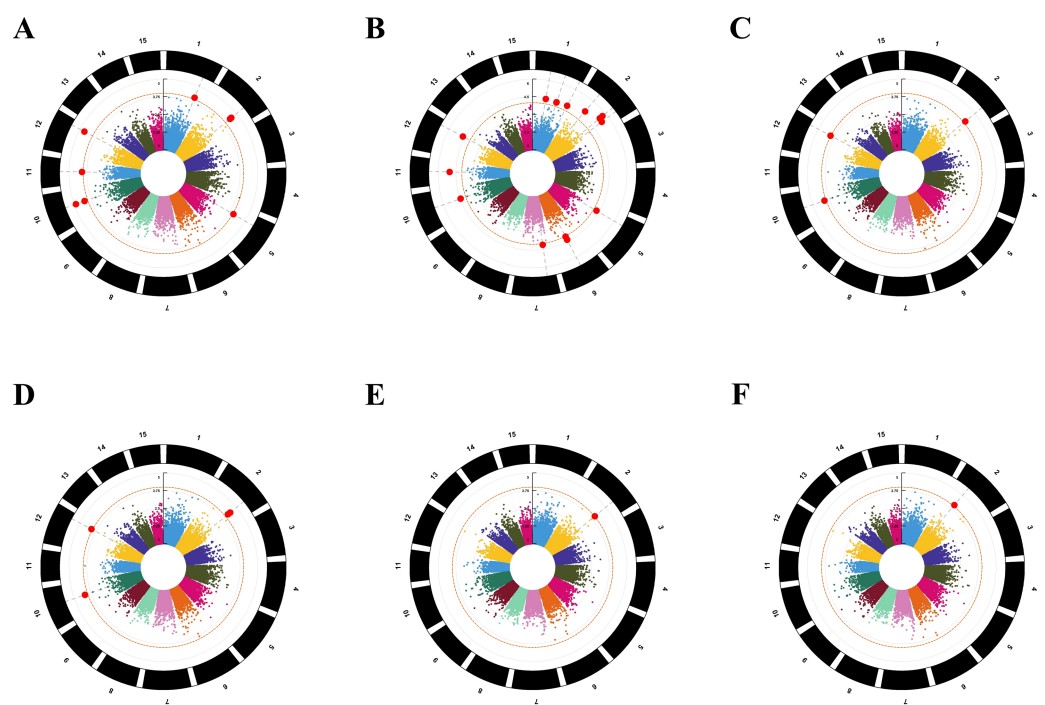

**Figure 4  Manhattan plots for the amino acid traits of tea plant by six models.** The orange dashed horizontal line indicates the significance threshold; the red solid line horizontal line indicates the extreme significance threshold. (A) GLM-P; (B) GLM-Q; (C) MLM-P+K; (D) MLM-Q+K; (E) CMLM-P+K; (F) CMLM-Q+K.

**Table 2  The optimal model of amino acid of 212 tea accessions was used for SNPs analysis.**

| Traits | Model | SNPs | Chromosome | Position | $-\text{Log}_{10}P$ | $R^2$(%) | Allele |
|--------|-------|------|------------|----------|---------|----------|--------|
| Aa | GLM(P) | S1_169435208 | 1 | 169435208 | 4.117 | 10.45% | G/A |
| | | S2_151915980 | 2 | 151915980 | 4.534 | 11.36% | G/A |
| | | S2_151915996 | 2 | 151915996 | 4.417 | 11.31% | T/C |
| | | S5_46409247 | 5 | 46409247 | 4.057 | 8.82% | T/C |
| | | S10_90937109 | 10 | 90937109 | 4.233 | 9.41% | G/C |
| | | S10_90937133 | 10 | 90937133 | 4.880 | 10.49% | C/T |
| | | S11_64920241 | 11 | 64920241 | 4.089 | 9.65% | C/T |
| | | S12_134931733 | 12 | 134931733 | 4.647 | 10.49% | A/G |

structure and genetic diversity analysis are of great significance for understanding the origin, evolution, and utilisation of crop germplasm. Meanwhile, it is also the basis for improving crop traits. Understanding the current level of crop diversity was one of the key factors for making the most effective use of crop germplasm for genetic improvement (*Lavanya, Srivastava & Ranade, 2008*; *Xia et al., 2020a*). *Zhao et al. (2022)* used 112,072 high-quality genotyping-by-sequencing to analyse the genetic diversity, principal components, phylogeny, population structure, and linkage disequilibrium and develop a core collection of 253 cultivated-type tea plant accessions from the Guizhou

**Table 3** Functional annotation of four candidate genes related to the amino acid levels.

| Traits | SNP locus | Chromosome | Candidate gene | Function annotation |
|---|---|---|---|---|
| Aa | S5_46409247 | 5 | TEA022785.1 | Ribokinase: involved in the pentose phosphate pathway |
| | S11_64916608 | 11 | TEA027937.1 | L-tryptophan–pyruvate aminotransferase: participate in lAA biosynthesis |
| | S11_64920241 | 11 | TEA013853.1 | DNA methylation factors: Methyltransferase activity |
| | S12_134931733 | 12 | TEA008491.1 | Xylan alpha-glucuronosyltransferase: may be involved in the substitutions of the xylan backbone in stem glucuronoxylan |

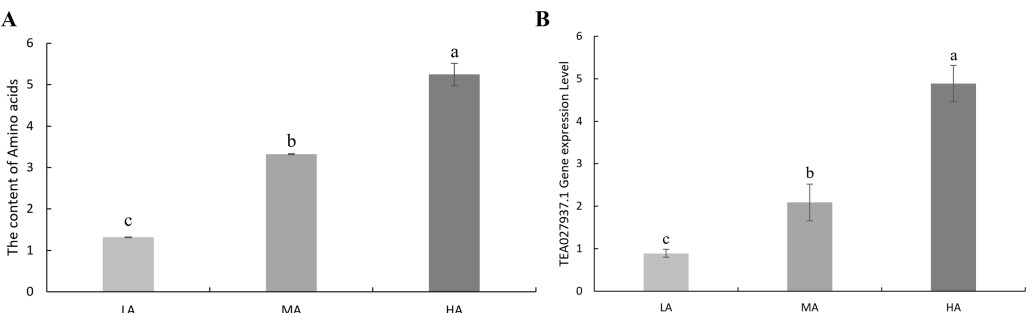

**Figure 5** **Amino acid content and expression levels of candidate genes.** (A) The content of amino acids in tea plants. (B) The expression level of *TEA027937.1* gene related to amino acids in tea plants. HA, leaves with high amino acid content; MA, leaves with moderate amino acid content; LA, Leaves with low amino acid content.

Plateau. *He et al. (2023)* analysed the genetic diversity, population structure, principal components, phylogeny, and linkage disequilibrium of 159 wild tea trees from different altitudes and different geological types in the Guizhou Plateau, and the results revealed the genetic diversity and geographical distribution characteristics of wild tea trees in the Guizhou Plateau. In order to understand the population structure and genetic relationship of 212 tea accessions, the population structure, the phylogenetic tree, and the genetic differentiation analysis were carried out. In this study, 212 tea accessions were divided into four subgroups. The phylogenetic tree also divides into four categories. Q1 and Q2 were mainly cultivars; Q3 and Q4 were mainly wild-type. Previous studies showed that after the domestication of species, due to factors such as geography and human selection, reproductive isolation occurred, and different populations, such as cultivated species and wild species, were gradually formed. However, some degree of gene flow still occurs between geographically close populations (*Wang et al., 2017*). Some studies have shown that gene flow and reproductive isolation may jointly affect the structure and traits of tea populations. Tea populations in different regions communicate through gene flow, while unique tea types are formed in a given region due to reproductive isolation (*Seehausen et al., 2014*; *Rabosky, 2016*; *Xia et al., 2020b*). For example, the tea tree was first discovered in China, but it is now an important evergreen crop grown in more than 50 countries around the world. Tea types with unique local characteristics have also been developed in different regions of China, such as West Lake Longjing, Yunnan Pu'er, and Wuyi Rock Tea. This

dynamic influence not only promotes the diversification of tea germplasm but also has an impact on the quality and adaptability of tea.

## Genome-wide association study of amino acids

Plant material in association studies should have a wide genetic diversity that maximises the capture of historical recombination events. In this study, the amino acid traits of 212 tea accessions were investigated, and the results showed that the amino acid traits of 212 tea plants had a wide range of variation. In addition, 78,819 high-quality SNPs from 212 tea accessions were retained after filtration, which was comparable to the number of 79,016 high-quality SNPs obtained from 415 Guizhou tea accessions reported previously, indicating that the 212 tea accessions from Guizhou had rich genetic diversity and could be used for GWAS (*Niu et al., 2019*). GWAS has emerged as a powerful approach for the identification of genetic regions underlying complex traits in crop plants (*Wu et al., 2017*). In sweet potato, a total of seven genes involved in sugar and starch metabolism in sweet potato were identified through genome-wide association (*Nie et al., 2023*). *Yang et al. (2022a)* presented a genome-wide association study revealing candidate genes related to stem diameter of cucumber (*Cucumis Sativus* Louis). *Li et al. (2021b)* revealed Four quantitative trait loci for germination rate were identified using a genome-wide association study during early germination in rice. Although GWAS has been extensively studied in major food crops, there are fewer studies on the use of GBS for GWAS for amino acid traits in tea compared to other crops. But the above research provided a good reference for the quality breeding of horticultural crops such as tea plants. As one of the birthplaces of tea, Guizhou is rich in wild tea germplasm resources. Tea is the most drunk beverage in the world and is processed from young leaves. As one of the main quality components of tea, amino acids play an important role in the flavor of tea. Mining candidate genes associated with amino acids can improve the abundance of key compounds associated with the flavor of tea, thereby improving the health benefits of tea plants. In this study, we employed genotyping sequencing-based GWAS to develop amino acid-related SNP loci and identify potential candidate genes. (GLM) (GLM-P and GLM-Q), (MLM) (MLM-P+K and MLM-Q+K), and (cMLM) (cMLM-P+K and cMLM-Q+K) were the six most common statistical models in GWAS. The Quantile–Quantile plots (Q–Q plots) represented the degree to which the model fits the data and were used to judge the deviation between the observed and expected values. The Manhattan diagram showed the correlation between the SNPs locus on each chromosome and the trait. We identified 17 amino acid-related SNPs under six GWAS models, of which eight ultimately significant SNPs were identified in one GWAS model determined by the best QQ plot. GLM is a general linear model, which has a wider range of applications and can screen more SNPs related to traits, but the accuracy is lower than that of MLM. MLM is a hybrid linear model. Compared with GLM, the model needs to add a genealogy matrix, *i.e.*, the kinship K matrix. On the basis of MLM, MLM is further optimised to obtain CMLM, which is called the compressed hybrid linear model. Since MLM correction is too strict and filters out some of the SNP markers that are really relevant, CMLM aims to redetect those false-negative SNP markers. It is important to note that all three models can add principal components (PCA) and population structure

(Q) to control for possible false positives for traits. In this study, GLM-P was selected according to the best QQ plot. Eight significant SNPs were associated with the GLM-P. Among the 8 significant SNPs, in addition to GLM-P, S2_151915980 was also found in GLM-Q, MLM-P+K, MLM-Q+K, and CMLM-P+K. S2_151915996 can also be found in GLM-Q and MLM-Q+K; S5_46409247 can also be found in GLM-Q; S10_90937133 can also be found in GLM-Q, MLM-P+K, and MLM-Q+K; S11_64920241 can also be found in GLM-Q; S12_134931733 can also be found in GLM-Q, MLM-P+K, and MLM-Q+K. Most of the SNPs screened by the model determined according to the best QQ diagram can coincide with the SNPs screened by other models, which proves that the selection of the best model is accurate.

## Prediction of candidate genes

LD decay distance is used to identify regions of potential candidate genes in GWAS. To confirm the beneficial alleles on each SNP peak associated with the seedless phenotype, *Zhang et al. (2017)* considered the gene model of the LD decay distance genomic region upstream and downstream of each SNP peak in the reference grape genome as a seedless candidate gene. Our results show that the attenuation distance is about 5 kb when $r^2$ drops to half. Based on LD attenuation, genes located within the 5 kb region surrounding trait-related SNPs were identified as potential candidate genes. We found an interesting phenomenon that when we extended the region to 50 kb and 100 kb, we found that we got the same four candidate genes as in the 5 kb range. Quantitative real-time polymerase chain reaction (qPCR) or reverse transcription qPCR (RT-qPCR) is a commonly used tool for gene expression analysis, detection, and quantification (*Foley, Leonard & Engel, 1993*). *Mahmoudzadeh et al. (2016)* established a simple and rapid method for extracting RNA from beef matrix to reduce the error caused by matrix effects on the experimental results. *Đermić et al. (2023)* have developed a new method for RT-qPCR using a specially modified primer in the reverse transcription step. All of these studies demonstrate that reverse transcription qPCR (RT-qPCR) can be used as a reliable and sensitive method for studying gene expression. We chose a kit that is easy to operate and has a strong matrix removal effect for the test, and the test is carried out in strict accordance with the requirements of all kits, so as to minimise the impact of the external environment on the test results. The expression levels of the above four genes were verified by RT-qPCR, and only the expression level of the *TEA027937.1* gene was significantly positively correlated with the amino acid content, which showed that the gene could be used as a candidate gene related to amino acid traits, and the function of the gene was L-tryptophan-pyruvate aminotransferase. L-tryptophan aminotransferase has been shown to be involved in auxin (IAA) biosynthesis. In plants, hormones derived from amino acid metabolic pathways, such as auxin (L-tryptophan) and ethylene (L-methionine), alter development and growth rates optimised for varying environmental conditions (*Santner & Estelle, 2009*; *Jaillais & Chory, 2010*). Indoleacetic acid (IAA) is the most studied naturally active auxin. There are two main pathways for the biosynthesis of IAA: the tryptophan-dependent pathway and the tryptophan-independent pathway (*Woodward & Bartel, 2005*). The indole-3-pyruvate (IPyA) pathway, a tryptophan (Trp)-dependent pathway, appears to be a major contributor to the formation of free

IAA (*Zhao, 2012*). L-tryptophan aminotransferase converts L-tryptophan and pyruvate to indole-3-pyruvic acid (IPA) and alanine (*Mashiguchi et al., 2011*; *Won et al., 2011*). This is the first step in the IPyA branch of the tryptophan-dependent pathway. Arabidopsis tryptophan aminotransferase (TAA1) is the first known major enzyme for the biosynthesis of indole-3-acetic acid (IAA) from l-tryptophan, which belongs to the TAA1-related (TAA1/TAR) tryptophan aminotransferase family and has been identified in several mutant screens (*Stepanova et al., 2008*; *Tao et al., 2008*; *Yamada et al., 2009*). Previous studies have revealed that TAA1/TAR protein transaminase not only plays an important role in IAA biosynthesis but also plays an important role in plant nitrogen cycling (*Le Deunff et al., 2019*). These reports confirm our finding that one potential candidate gene identified in this study may be related to amino acids in tea plants, which will be beneficial for the development of molecular markers and MAS breeding in the future.

Amino acid content is a quantitative trait. Quantitative traits in plants are generally controlled by micro-effect polygenes. However, factors such as population size, recombination events, and genetic heterogeneity can affect the ability of QTLs to localise to identify loci or genes (*Price, 2006*). Omics techniques, including metabolomics, transcriptomics, proteomics, and ionomics, have contributed greatly to the identification and characterisation of genes, proteins, metabolites, and ions involved in the signalling pathways of interest (*Budak et al., 2015*). Transcriptome sequencing is an effective strategy to explore gene expression dynamics. In addition, metabolomics can be considered a complementary technique to transcriptomics (*Lake et al., 2017*). With the rapid development of bioinformatics and biostatistics tools, transcriptomics and metabolomics-based can correlate a large number of genes and metabolites to obtain a complete overview (*Wu et al., 2019*). For example, recent advances in metabolomics and transcriptomics techniques have greatly enhanced our understanding of flavonoid biosynthesis pathways and their regulatory networks (*Wu et al., 2024*). In tea plants, we can also combine multi-omics technology to conduct a comprehensive analysis of amino acid metabolism pathways in tea plants to further clarify the genetic basis of amino acids in tea plants.

## CONCLUSIONS

This study demonstrates that it is feasible to generate large-scale SNP markers using the GBS strategy to analyse the genetic diversity and GWAS of amino acids in tea plants. Based on 78,819 SNPs developed by GBS, the genetic diversity of 212 tea accessions from Guizhou Province was analyzed. All 212 tea germplasms showed high genetic diversity. We used GWAS to identify a potential candidate gene associated with amino acid. It was found that this candidate gene was closely related to the auxin anabolic pathway. However, the auxin synthesis pathway is regulated by the amino acid metabolism pathway. In addition, the amino acid content of tea plants may be affected by external factors, so phenotypic data needs to be collected for several consecutive years to confirm this possibility. In conclusion, the results of this study can provide a reference for the future exploration of candidate genes for complex traits in tea plants, which is helpful for the breeding of excellent quality traits in tea plants.

## ACKNOWLEDGEMENTS

We would like to thank all the participants for their help in completing this study together.

### Funding

This work was supported by the National Science Foundation of China (32060700), National Guidance Foundation for Local Science and Technology Development of China ([2023] 009), Project on Guiyang Science and Technology Plan (Construction Technology Contract [2023] 48-21), Wangmo Eight Step Tea: Integration and Demonstration of Quality and Efficiency Enhancement Technologies (2021YFD1100307), the Science and Technology Plan Project of Guizhou Province, in RP China ([2023] General 480) and Youth Science and Technology Talent Project of Education Department of Guizhou Province (KY 2022-153). The funders had no role in study design, data collection and analysis, decision to publish, or preparation of the manuscript.

### Grant Disclosures

The following grant information was disclosed by the authors:
National Science Foundation of China: 32060700.
National Guidance Foundation for Local Science and Technology Development of China: [2023] 009.
Project on Guiyang Science and Technology Plan: Construction Technology Contract: [2023] 48-21.
Wangmo Eight Step Tea: Integration and Demonstration of Quality and Efficiency Enhancement Technologies: 2021YFD1100307.
Science and Technology Plan Project of Guizhou Province, in RP China: [2023] General 480.
Youth Science and Technology Talent Project of Education Department of Guizhou Province: KY 2022-153.

### Competing Interests

The authors declare there are no competing interests.

### Author Contributions

- Qidi Wu conceived and designed the experiments, performed the experiments, analyzed the data, prepared figures and/or tables, authored or reviewed drafts of the article, and approved the final draft.
- Li Song conceived and designed the experiments, authored or reviewed drafts of the article, and approved the final draft.
- Dingchen Bai analyzed the data, prepared figures and/or tables, and approved the final draft.
- Yihan Wang analyzed the data, prepared figures and/or tables, and approved the final draft.

- Yuting OuYang performed the experiments, analyzed the data, prepared figures and/or tables, and approved the final draft.
- Kaixin Rao performed the experiments, analyzed the data, prepared figures and/or tables, and approved the final draft.
- Qinfei Song conceived and designed the experiments, authored or reviewed drafts of the article, and approved the final draft.
- Juanying Wang conceived and designed the experiments, authored or reviewed drafts of the article, and approved the final draft.
- Suzhen Niu conceived and designed the experiments, authored or reviewed drafts of the article, and approved the final draft.
- Yujie Ai conceived and designed the experiments, authored or reviewed drafts of the article, and approved the final draft.

## Data Availability

The data is available in the Supplemental Files.

## Supplemental Information

Supplemental information for this article can be found online at http://dx.doi.org/10.7717/peerj.18969#supplemental-information.

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
