# Peer review of "Genome-wide association studies reveal potential candidate genes associated with amino acid in tea plants"

_PeerJ, doi:10.7717/peerj.18969_

## Round 0.1 · original submission · Major Revisions

Dear Authors

The manuscript cannot be accepted for publication in its current form. It needs a major revision before publication. The authors are invited to revise the paper considering all the suggestions made by the reviewers. Please note that the requested changes are required for publication.

With Thanks

Reviewer 1 ·

Basic reporting

The article presents a detailed investigation into the genetic basis of amino acid content in tea plants, utilizing genome-wide association studies (GWAS) to identify potential candidate genes. The research addresses a relevant and timely topic, as amino acid profiles are crucial for tea quality and nutritional value. While the study offers valuable insights, certain sections would benefit from refinement to improve clarity, methodological rigor, and overall impact.
In introduction, while the specific benefits of Guizhou’s location are mentioned, the rationale for studying this region’s germplasm should be clarified. For example, how does Guizhou’s low latitude and high altitude influence tea plant genetics or amino acid synthesis?
The introduction briefly hints at the need for research on amino acid-associated gene loci in tea but does not clearly define the existing knowledge gap. Expanding on why identifying these genes is significant for genetic improvement (e.g., flavor enhancement, resilience) would strengthen the hypothesis and scientific motivation.
The relevance of GWAS and GBS techniques in this study could be explained in more depth, showing why these techniques are well-suited for this genetic analysis and how they improve upon previous research methods.
Please provide more latest literature on biochemical traits or sensory characteristics related to amino acids in tea.
Define the knowledge gap, such as the lack of understanding of specific amino acid genes in tea and how identifying these can assist in breeding programs.
Establish a hypothesis about gene loci for amino acids and clarify objectives with focus on both genetic diversity and trait improvement.
In material and methods section please provide information on the timing, handling, and environmental conditions during sample collection. Describe the storage protocols, such as drying method (air or freeze-dried).
The methods suggest that Guizhou tea germplasm was chosen for its genetic uniqueness, which aligns well with the research gap related to under-explored genetic diversity in this region. However, further justification for including two non-Guizhou samples from other provinces should be explained.
Describing the use of an ultraviolet spectrophotometer to measure amino acids based on the national standard (GB/T8314-2013) is appropriate. However, to enable reproducibility, specific wavelength settings, sample preparation procedures, and measurement conditions should be described, including any calibration protocols.
The software used (Origin2021, IBM SPSS) is appropriate, but more detail is needed, specify which statistical tests were used for mean, standard deviation, and coefficient of variation. Please provide this information in the end of materials and methods section under separate heading e.g., Statistical analyses
The SNP filtering and removal processes could be clarified with details on software and parameters, which would ensure consistency in replicating the analysis.
Specifying cross-validation (CV) iterations for population structure study. The LD decay analysis description should include details on window size and R² threshold.
More detail on the interpretation of results (e.g., expected Fst threshold values) and rationale for these analyses would improve the section’s sufficiency.
Specify software version numbers, input parameter details, and quality control steps for GWAS data (e.g., missing data handling).
The choice of a -Log10(P) threshold of 4.0 to identify significant SNP loci is suitable, though a justification or reference for this threshold would be beneficial.
To identifying candidate genes within 5 kb of significant SNP loci include genome assembly details, specific search parameters, and database versions.
Describe cDNA synthesis reaction conditions (e.g., temperature and time), primer sequences for target and reference genes, and reaction conditions for RT-qPCR (e.g., cycle number, annealing temperature).
Specify the statistical approach to validate the data (e.g., ANOVA for biological replicates) would support technical rigor.
In results section clearly justify why K=3K = 3K=3 was chosen over other values, despite similar CV error values for K=2K = 2K=2 and K=3K = 3K=3. Including more quantitative justification, such as visual or statistical differentiation at these levels, could enhance clarity.
Interpret how higher He or Fis values relate to genetic structure and diversity within specific subgroups.
The statement that Tajima's D>0D > 0D>0 reflects bottleneck effects should be clarified with reference to specific allelic frequency patterns observed here, which would solidify the argument.
Clarify how model selection was validated or justified, beyond the QQ plots.
The discussion about gene flow and reproductive isolation is a key point that could be enriched by briefly explaining how these dynamics affect tea population structure and traits across regions.
The introduction to GWAS in crop plants is good but could briefly tie back to why amino acids are valuable traits in tea breeding. Explaining why amino acid composition in tea leaves matters—perhaps in terms of flavor profile, nutritional value, or stress response—would clarify the relevance of these findings.
The section describes different models (GLM, MLM, CMLM) used to identify significant SNPs. While this is technically accurate, the explanation would benefit from a sentence about why each model’s characteristics matter in this study. For instance, why is GLM-P chosen based on the QQ plot, and what does this mean for the reliability of identified SNPs?
Rather than simply listing which models each SNP appears in, it would be helpful to highlight any patterns or potential implications regarding amino acid traits in tea plants.
The LD decay distance discussion is clear and well-supported with reference to prior studies on candidate gene regions. The section provides a logical rationale for why a 5 kb distance was selected for identifying candidate genes.
The finding of TEA027937.1 as a candidate gene associated with amino acid content, particularly its role in the auxin biosynthetic pathway, is presented well. However, further depth is needed on why this pathway might be significant for amino acid synthesis or other physiological traits in tea. Expanding briefly on how auxin biosynthesis and amino acid metabolism might interact to affect growth or quality traits in tea would make this finding more impactful.
Discuss how does genetic diversity potentially impact the traits explored in the GWAS study?
While the conclusion section highlights findings on genetic diversity and SNPs related to amino acid synthesis, it would benefit from contextualizing these findings in terms of broader implications for tea breeding, conservation, and potential applications in developing tea varieties with improved amino acid profiles. The statement that tea plants could not be “completely distinguished” based on geographical origin is somewhat vague. It would help to specify if this result implies gene flow across regions, similar environmental adaptation, or limited genetic drift. The conclusion could benefit from a brief mention of the limitations of the study and further future research recommendations.
Specific comments:
Line 25-26: The phrase "main components that determine the taste of tea soup" could be revised to "primary components determining the taste of tea," as "tea soup" is an uncommon term.
Line 37: "There may be one potential gene that are important" should be "there may be one potential gene that is important.
Line 49: While the introduction presents relevant terms, some expressions, like "fishy smell, seaweed flavor, fresh sweetness," lack technical refinement and could be improved with descriptors that better reflect amino acids’ impact on flavor chemistry. Instead of repetitive phrases like “seaweed flavor” twice, the terms could be more concise and descriptive, like “characteristic umami profiles."
Line 109: "Use Origin2021 software to draw a box graph..." This should be revised to "Origin2021 software was used to create box plots of amino acid content to identify and exclude outliers."
Line 110: "And use IBM SPSS Statistics..." This should be changed to "IBM SPSS Statistics (v. 26) was used to analyze the range, mean, standard deviation, and coefficient of variation of traits."
Line 149: "We select the best model for the trait to screen for significant SNP loci and search..." should be "The best model for the trait was selected to screen for significant SNP loci and to search..."
Line 156: "9 tea materials (including three materials..." should be written out as "Nine tea materials (including three with high amino acid content..."
Line 298: The phrase "Population structure and genetic diversity analysis was great significance" should be revised to "Population structure and genetic diversity analysis are of great significance."
Line 370: Consider rephrasing "according to their geographical origin" as "by their geographical origin."

Experimental design

The experimental design lacks critical details that are essential for assessing the validity and replicability of the study. The authors provide only limited information. However, to properly evaluate the rigor of the experimental setup, more information is necessary, including the design type (e.g., randomized block or split-plot), plot size, replication, and environmental controls. Details on the randomization of plant samples and measures to mitigate external environmental variability, as well as a clear rationale for the selection and number of plants used, are needed. Without this information, it is challenging to validate the reliability of the data or the reproducibility of the findings.

Validity of the findings

Validity of the findings can be decided on the basis of experimental design.

Reviewer 2 ·

Basic reporting

The literature references should have more on explanation of the results.

Experimental design

OK

Validity of the findings

Need to have more perspective on how to enhance the reliability of GWAS results in metabolite pathway level.

Additional comments

Review peerj-107558

The work contains some useful information. However, GWAS has been applied in various plants and widely reported. Therefore, the authors need to enhance the elaboration of the novelty of this report. Secondly, GWAS analysis revealed potential genes associated with. However, gene level might be able to be reflected in metabolites, the metabolomics level. More perspective is needed on elaborating how to enhance the results from other approaches as future research.

Only gene level is hard to elaborate the real metabolites in amino acid metabolic pathways as many genes can develop adaptive responses under different scenarios. To integrate with other omics like metabolomics might be helpful to better elucidate the adaptive responses of the genes or metabolic pathways. The perspective section could be enhanced. The authors can search references with key topic like ‘integrated metabolomics and transcriptomics reveal adaptive responses’ to discuss it further.

For RT-qPCR analysis, the results indicated that matrix compounds removal was pretty successful for achieving good assay accuracy. However, the matrix effect was widely existed for such delicate analysis. Thus, in-depth discussion is needed. The authors are suggested to search database like Web of Science with matrix compounds removal (Title) AND assay (Title) to get reference to discuss it further.

The genes analysis should further be linked to different metabolic pathways associated with thus to be understood logically. The authors can search database like Web of Science with food metabolomics (Title) AND quality analysis (Title) to get reference to enhance the discussion.

Fig. 2D and Fig. 5: Use some letters or symbols to indicate the relationship among the four groups. For instance, whether there were significant differences among the groups.

Reviewer 3 ·

Basic reporting

Wu et al. Conducted a GWAS in tea plants using GBS and amino acid data. The analysis are correct. I also consider this work should be published. However, first, a revision on English grammar is strongly suggested.
• Abstract. Include the traits that were evaluated.
• Keywords: Don’t repeat the words that are in the title of the manuscript
• Line 93, 156, and more. Don´t begin the sentence with a number
• Line 93. Please indicate the relation between the 212 and 415 germplasm resources
• Line 108, 109, 198, others. Sentences are not clear.
• Line 121. Indicate version of ADMIXTURE.
• Line 128. Change “is” to “was”
• Line 125, 127, 128 and more. Add citation and version of software
• Line 130. Sentence is not clear. Indicate what Ho, MAF, etc, mean
• Line 131, 132, Indicate what Fst, Pi, etc, mean
• References. Some scientific names are not in italics.
• Fig 1A. What does the triangle mean?
• Fig 1B. Indicate what color blue and orange mean
• Fig 1C. Indicate % of variance for each PC
• Molecular and amino acid data should be uploaded into a repository.
• Suppl material is fine

Experimental design

It is OK

Validity of the findings

It is OK

---

## Round 0.2 · accepted · Accept

Dear Authors,

I am pleased to inform you that the manuscript has improved after the last revision and can be accepted for publication.

Congratulations on accepting your manuscript, and thank you for your interest in submitting your work to PeerJ.

With Thanks

Reviewer 1 ·

Basic reporting

I am pleased to recommend acceptance of the revised manuscript. The authors have addressed all the concerns and suggestions raised in my previous review, and the manuscript has improved significantly.

Experimental design

Experimental design is sufficiently detailed for reproducibility.

Validity of the findings

Results seems original and valid.

Reviewer 2 ·

Basic reporting

OK

Experimental design

OK

Validity of the findings

OK

Additional comments

The authors have addressed the questions quite well. The quality of the revised manuscript has been improved significantly. However, the authors need to double check some format and spelling issues during proofreading. For instance, Line 599, the volume and issue information should be provided as complete citation format is available for this article.